# Rational design of isostructural 2D porphyrin-based covalent organic frameworks for tunable photocatalytic hydrogen evolution

Rufan Chen[1,5], Yang Wang[2,3,5], Yuan Ma [4,5], Arindam Mal [1], Xiao-Ya Gao[2,3], Lei Gao [4], Lijie Qiao[4], Xu-Bing Li[2,3✉], Li-Zhu Wu [2,3✉] & Cheng Wang [1✉]

Covalent organic frameworks have recently gained increasing attention in photocatalytic hydrogen generation from water. However, their structure-property-activity relationship, which should be beneficial for the structural design, is still far-away explored. Herein, we report the designed synthesis of four isostructural porphyrinic two-dimensional covalent organic frameworks (MPor-DETH-COF, M = H$_2$, Co, Ni, Zn) and their photocatalytic activity in hydrogen generation. Our results clearly show that all four covalent organic frameworks adopt AA stacking structures, with high crystallinity and large surface area. Interestingly, the incorporation of different transition metals into the porphyrin rings can rationally tune the photocatalytic hydrogen evolution rate of corresponding covalent organic frameworks, with the order of CoPor-DETH-COF < H$_2$Por-DETH-COF < NiPor-DETH-COF < ZnPor-DETH-COF. Based on the detailed experiments and calculations, this tunable performance can be mainly explained by their tailored charge-carrier dynamics via molecular engineering. This study not only represents a simple and effective way for efficient tuning of the photocatalytic hydrogen evolution activities of covalent organic frameworks at molecular level, but also provides valuable insight on the structure design of covalent organic frameworks for better photocatalysis.

[1] Sauvage Center for Molecular Sciences and Key Laboratory of Biomedical Polymers (Ministry of Education), College of Chemistry and Molecular Sciences, Wuhan University, Wuhan, China. [2] Key Laboratory of Photochemical Conversion and Optoelectronic Materials, Technical Institute of Physics and Chemistry, Chinese Academy of Sciences, Beijing, China. [3] School of Future Technology, University of Chinese Academy of Sciences, Beijing, China. [4] Beijing Advanced Innovation Center for Materials Genome Engineering, Institute for Advanced Materials and Technology, University of Science and Technology Beijing, Beijing, China. [5] These authors contributed equally: Rufan Chen, Yang Wang, Yuan Ma. ✉email: lixubing@mail.ipc.ac.cn; lzwu@mail.ipc.ac.cn; chengwang@whu.edu.cn

Covalent organic frameworks (COFs) are a novel class of porous crystalline polymer that enables the precise integration of molecular building blocks into extended two-dimensional or three-dimensional (2D or 3D) structures through covalent bonds[1–5]. Owing to their low density, high porosity, structural periodicity, and modular functionality, COFs have gained intensive attention and found promising applications in gas adsorption and separation[6–10], catalysis[11–14], sensing[15–18], optoelectronics[19–22], and energy storage[23–26]. From the structural viewpoint, the most important feature of 2D COFs differing from their 3D analogues and most organic systems is that they can offer a unique platform for constructing periodic columnar π arrays[27]. Accordingly, 2D COFs possess unique pre-organized transport of long-lived photoexcited states and show high charge carrier mobility, which will allow them to work as effective heterogeneous photocatalysts. In addition, the crystalline nature of 2D COFs can facilitate the establishment of structure–property–activity relationship and thus providing insights into photocatalytic processes. Therefore, 2D COFs have received growing interests in photocatalysis over the past few years, ranging from chemical transformation[28–33] to solar fuel production[34–48].

Among all these tested systems, photocatalytic hydrogen evolution reaction (HER) from water is regarded as one of the most attractive ways to meet the increasing demands of clean and sustainable energy[49]. In 2014, Lotsch and co-workers reported the first example of utilizing 2D COF to produce $H_2$ in the presence of metallic platinum under visible light irradiation[34]. Since this pioneer work, several 2D COFs bearing different photoelectric units have been successfully constructed and found interesting potential in photocatalytic hydrogen evolution[34–43]. However, although continuing efforts are going on developing new 2D COFs for photocatalytic HER, the rational tuning of their structures and photophysical properties for maximizing the hydrogen evolution efficiency still needs to be further clarified. In an initial study, Lotsch et al. reported several 2D COFs with different numbers of nitrogen atoms in the central phenyl ring[35], which showed controllable photocatalytic hydrogen evolution efficiencies. Unfortunately, as the tailoring of their photocatalytic performance lies on a multitude of variables (e.g., crystallinity, optoelectronic factors, etc.), it is very difficult to determine the individual contribution that is required for further modification. Therefore, it is highly demanded to construct isostructural 2D COFs with tunable optoelectronic properties and further explore their structure–property–activity relationship in photocatalytic HER from a molecular level.

Porphyrin and its derivatives, a kind of conjugated π-electron macrocycles with unique photophysical and redox properties[50,51], have been used to construct 2D COFs for heterogeneous photocatalysis[28,29,46–48]. In principle, the incorporation of different metal ions into porphyrin units may rationally tune their photophysical and electronic properties, which can thus affect the photocatalytic activity of corresponding COFs. With this consideration in mind, we report herein the synthesis and characterization of four isostructural hydrazone-linked 2D porphyrinic COFs (Fig. 1), named as MPor-DETH-COF (M = $H_2$, Co, Ni, Zn). Our results clearly demonstrate that these four COFs have high crystallinity and surface area, and the incorporation of different transition metal ions into porphyrin rings apparently influences the charge-carrier dynamics properties of corresponding COFs. When irradiated with visible light in the presence of $H_2PtCl_6$ and triethanolamine (TEOA), all MPor-DETH-COFs can continually produce hydrogen from water while retaining the framework. More importantly, these four COFs show rationally tunable activity toward photocatalytic hydrogen evolution with the order of CoPor-DETH-COF (25 μmol $g^{-1}$ $h^{-1}$) < $H_2$Por-DETH-COF (80 μmol $g^{-1}$ $h^{-1}$) < NiPor-DETH-COF (211 μmol $g^{-1}$ $h^{-1}$) < ZnPor-DETH-COF (413 μmol $g^{-1}$ $h^{-1}$), which can be mainly explained by their tailored charge-carrier dynamics via molecular engineering.

## Results

**COF synthesis and characterization.** In order to construct stable porphyrin-based 2D COFs for photocatalytic hydrogen evolution from water, we designed and synthesized porphyrinic aldehydes p-MPor-CHO (M = $H_2$, Co, Ni, and Zn), which could react with 2,5-diethoxyterephthalohydrazide (DETH) to form the designed isostructural MPor-DETH-COF (M = $H_2$, Co, Ni, and Zn) through condensation reaction (Fig. 1). It should be mentioned here, as the solubility and reactivity of these porphyrinic aldehydes are different, the reaction conditions need to be optimized to obtain high crystalline COFs. Generally, the condensation reaction was performed in a mixed solvent of 1,2-dichlorobenzene, butanol, and aqueous acetic acid at 120 °C, but the ratio of the solvents, the concentration and the amount of acetic acid and the reaction time were optimized for each COF synthesis. The chemical structure of these COFs was then assessed by a combination of Fourier transform infrared (FT-IR) spectroscopy and solid-state nuclear magnetic resonance (ssNMR) spectroscopy. From the FT-IR spectra, all MPor-DETH-COFs showed characteristic stretching vibration bands of C=N bond at 1670–1660 $cm^{-1}$ (Supplementary Fig. 4), indicating the formation of hydrazone linkage[52]. Furthermore, all of their ssNMR spectra showed a signal at ~162 ppm (Supplementary Figs. 5−7), confirming again the formation of hydrazone bond. From thermogravimetric analysis (Supplementary Figs. 8−11), all four COFs were thermally stable up to 350 °C.

The powder X-ray diffraction (PXRD) experiment was performed to elucidate the crystalline nature of these COFs. As shown in Fig. 2e−h, all MPor-DETH-COFs showed two main diffraction peaks at 3.0° and 6.1°, corresponding to (110) and (220) crystal facets, respectively. Detailed crystal model was then simulated using *Materials Studio* software package (see Supplementary section 4 for details) and the unit cell parameters were optimized according to density-functional tight-binding (DFTB) calculations. Obviously, the calculated diffraction patterns of AA eclipsed stacking model matched well with the experimental PXRD patterns, suggesting that all MPor-DETH-COFs adopted AA layer stacking structure. Furthermore, the Pawley refinement for these COFs yielded PXRD patterns (Fig. 2e−h) which were in good agreement with the experimentally observed data, as evidenced by the negligible difference. The detailed crystal structure information, including the unit cell parameters, could be found in Supplementary section 4. We also characterized the porosity of all MPor-DETH-COFs by nitrogen sorption isotherms measurement at 77 K. As shown in Fig. 2i−l, all of these COFs exhibited typical type-IV sorption isotherm curves, which is a characteristic evidence for mesoporous structures. The Brunauer–Emmett–Teller (BET) surface areas were calculated to be 826 $m^2$ $g^{-1}$, 942 $m^2$ $g^{-1}$, 773 $m^2$ $g^{-1}$, and 1020 $m^2$ $g^{-1}$ for $H_2$Por-DETH-COF, CoPor-DETH-COF, NiPor-DETH-COF and ZnPor-DETH-COF, respectively. Evaluation of the isotherms of all these COFs using quenched solid density functional theory (QSDFT) showed a main peak centered at around 2.4 nm, which agreed well with the calculated pore size (2.4 nm). To gain more insight of the microstructures of these four COFs, scanning electron microscopy (SEM) and high-resolution transmission electron microscopy (HR-TEM) were also performed. SEM images showed that all MPor-DETH-COFs acquired the same morphology of stacked sheets (Supplementary Fig. 12), where the corresponding HR-TEM images confirmed the layered stacking structure (Supplementary Fig. 13).

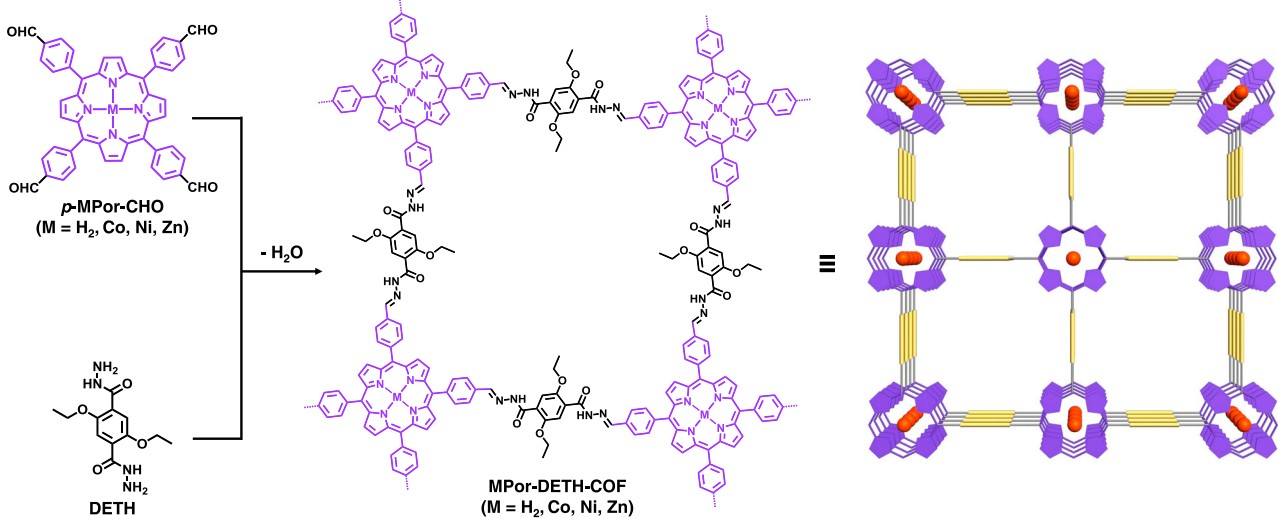

**Fig. 1 Chemical structure.** Schematic representation of the synthesis of MPor-DETH-COFs.

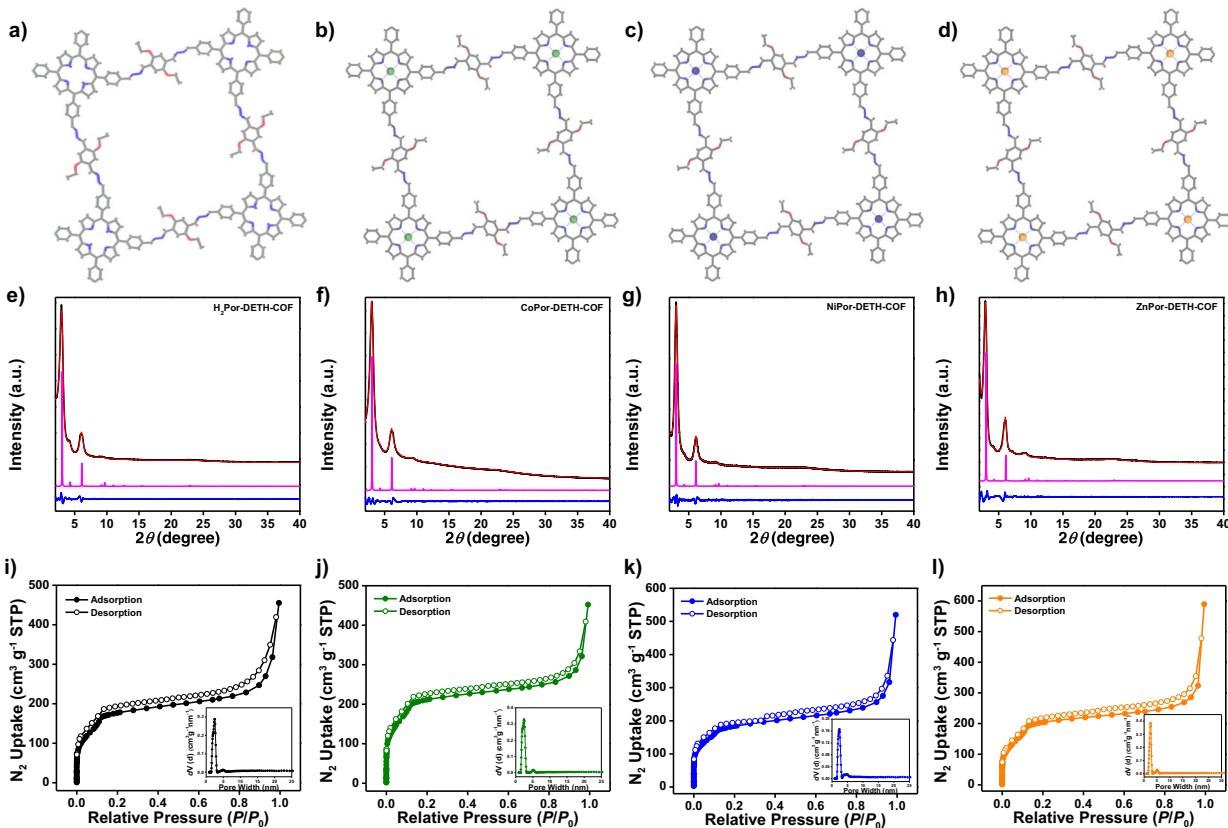

**Fig. 2 Structural characterization of MPor-DETH-COF.** Eclipsed AA stacking structures for **a** H$_2$Por-DETH-COF, **b** CoPor-DETH-COF, **c** NiPor-DETH-COF, and **d** ZnPor-DETH-COF. C, gray; O, red; N, blue; Co, dark green; Ni, navy blue; Zn, orange; H atoms are omitted. PXRD patterns of **e** H$_2$Por-DETH-COF, **f** CoPor-DETH-COF, **g** NiPor-DETH-COF, and **h** ZnPor-DETH-COF with the experimental profiles in black, Pawley-refined in red, predicted in magenta, and the differences in blue. N$_2$ sorption isotherms of **i** H$_2$Por-DETH-COF, **j** CoPor-DETH-COF, **k** NiPor-DETH-COF, and **l** ZnPor-DETH-COF. Insets: pore size distributions.

**Optical and electronic properties**. UV–Vis diffuse reflection absorption spectra of all MPor-DETH-COFs were first recorded (Fig. 3a). Compared with their precursors *p*-MPor-CHO (Supplementary Figs. 18 and 19), all the four COFs showed broader absorption edge with an apparent redshift up to ~660–700 nm, due to the formation of 2D extended networks. Moreover, all MPor-DETH-COFs showed similar absorption spectrum below 500 nm (B band absorption of porphrins), indicating the

incorporation of metal ions (Co$^{2+}$, Ni$^{2+}$, and Zn$^{2+}$) only caused weak perturbations to the π-orbitals of porphyrin ring. By using *Tauc* plot, the optical band gaps of H$_2$Por-DETH-COF, CoPor-DETH-COF, NiPor-DETH-COF, and ZnPor-DETH-COF were calculated to be 1.77, 1.88, 1.82, and 1.88 eV, respectively (Supplementary Fig. 20). As these values are in a very narrow range (1.77–1.88 eV), the different metalation of porphyrin precursors cannot obviously influence the optical band gaps of

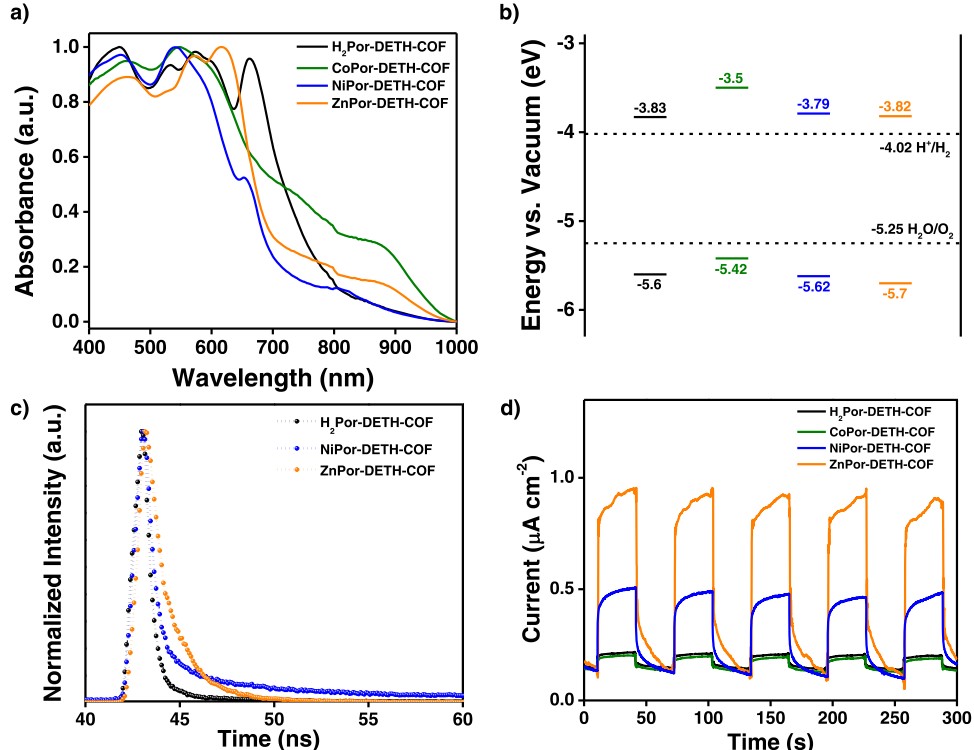

**Fig. 3 Optoelectronic properties of MPor-DETH-COF. a** UV–vis diffuse reflection absorption spectra of H$_2$Por-DETH-COF, CoPor-DETH-COF, NiPor-DETH-COF and ZnPor-DETH-COF. **b** Frontier orbital energies of all four COFs compared to the redox potentials of water. **c** The PL lifetime decay tests for all four COFs. Samples were excited with a $\lambda_{ex}$ = 405 nm laser and emission was measured at 682 nm for H$_2$Por-DETH-COF and ZnPor-DETH-COF, and 660 nm for NiPor-DETH-COF. **d** Photocurrent generation of all four COFs coated on an indium-tin-oxide electrode as a working electrode in a three electrode CV setup upon light on-off switching.

corresponding COFs. Moreover, cyclic voltammetry (CV) spectroscopy of these four COFs was performed to study their electronic bands (Supplementary Fig. 21). Accordingly, their highest occupied molecular orbital (HOMO) and lowest unoccupied molecular orbital (LUMO) were calculated. As shown in Fig. 3b, the LUMO energy levels were calculated to be about −3.8 eV for all four COFs except CoPor-DETH-COF (−3.5 eV), while the HOMO levels were −5.60, −5.42, −5.62, and −5.70 eV for H$_2$Por-DETH-COF, CoPor-DETH-COF, NiPor-DETH-COF, and ZnPor-DETH-COF, respectively (Supplementary Table 5). In addition, the valence band X-ray photoelectron spectroscopy (XPS) analysis of these four COFs exhibited comparable results (Supplementary Table 6). Obviously, these experimental orbitals locate at a very suitable scope for photocatalytic HER from water.

We then carry out the time-correlated single-photon counting (TCSPC) of all MPor-DETH-COFs to estimate their excited-state lifetimes in solid state, which are related to the carrier separation dynamics[42,53] arising from the π–π* transitions of the four COFs in solid state[54]. As shown in Fig. 3c, ZnPor-DETH-COF showed the longest emission lifetime while H$_2$Por-DETH-COF has the shortest value in the nanosecond range. However, for CoPor-DETH-COF, it is essentially non-emissive (Supplementary Fig. 24). According to literature[55], CoPor-DETH-COF should have an excited-state lifetime of perhaps several picoseconds. Therefore, the amplitude-weighted average lifetimes of these four COFs follow the order of ZnPor-DETH-COF > NiPor-DETH-COF > H$_2$Por-DETH-COF > CoPor-DETH-COF. Accordingly, ZnPor-DETH-COF has the most favorable excited-state charge separation, which may in turn be good for the utilization of excited electrons and holes in corresponding photoredox reactions. We further conducted photocurrent tests for all

MPor-DETH-COFs to evaluate their photoelectric responses, by coating them on indium-tin oxide (ITO) substrates under same conditions (i.e., same amount, identical electrode area, etc.). The chopped photocurrent–time (I–t) curves of these COFs based photoelectrodes were presented in Fig. 3d, and their photocurrent responses also showed an order of ZnPor-DETH-COF > NiPor-DETH-COF > H$_2$Por-DETH-COF > CoPor-DETH-COF. This result is consistent with the trend of emission decays, which confirms again the most efficient charge carrier transport in ZnPor-DETH-COF. Therefore, the incorporation of different metal ions into porphyrin precursors can influence the charge-carrier dynamics of the resulting COFs.

**Photocatalytic hydrogen evolution.** Encouraging by above results, we then evaluated the photocatalytic performance of all MPor-DETH-COFs toward hydrogen evolution from water under visible light irradiation (Xe-lamp 300 W, $\lambda$ > 400 nm) in the presence of H$_2$PtCl$_6$ and triethanolamine (TEOA), where H$_2$PtCl$_6$ was employed as the precursor of co-catalysts and TEOA worked as the sacrificial reagent. Control experiments confirmed that visible light, TEOA, H$_2$PtCl$_6$, and MPor-DETH-COFs were indispensable for effective hydrogen generation (Supplementary Fig. 27a–d). Moreover, H$_2$Por-DETH-COF could not produce hydrogen gas when the incident wavelength was above 500 nm (Supplementary Fig. 27e), indicating that B-band absorption of porphyrin ring mainly contributes to the photoredox reactions. Notably, all four COFs could constantly evolve hydrogen gas during 10 h light irradiation (Fig. 4a and Supplementary Fig. 28). The average rates of hydrogen evolution were quantified as 80 μmol g$^{-1}$ h$^{-1}$, 25 μmol g$^{-1}$ h$^{-1}$, 211 μmol g$^{-1}$ h$^{-1}$, and 413 μmol g$^{-1}$ h$^{-1}$ for H$_2$Por-DETH-COF, CoPor-DETH-COF,

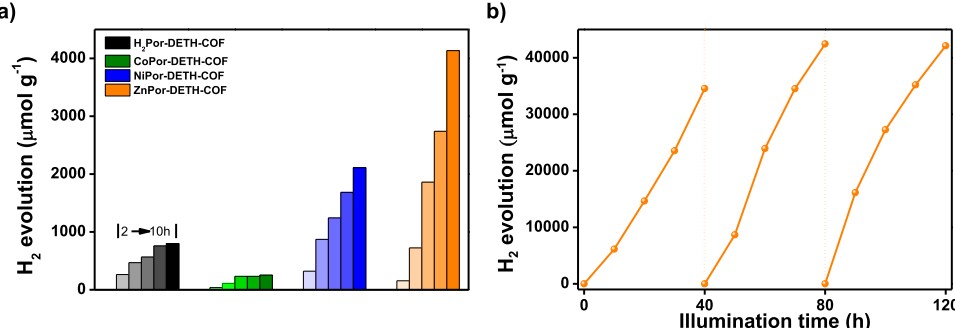

**Fig. 4 Hydrogen production rate. a** Time dependent $H_2$ photogeneration using visible light for $H_2$Por-DETH-COF, CoPor-DETH-COF, NiPor-DETH-COF and ZnPor-DETH-COF (2.5 µg catalyst in 5 mL phosphate buffer solution, 2.5 µL (8 wt% $H_2PtCl_6$), 50 µL TEOA, $\lambda > 400$ nm 300 W Xe lamp). **b** Long-term $H_2$ production using visible light for ZnPor-DETH-COF for 120 h (recycled every 40 h for three times).

NiPor-DETH-COF and ZnPor-DETH-COF, respectively. It should be emphasized that, this trend of hydrogen evolution efficiency matches well with the transient emission decays and chopped photocurrent tests of the four COFs, indicating the important factor of charge-carrier dynamics. The apparent quantum efficiency (AQE) of ZnPor-DETH-COF towards photocatalytic hydrogen evolution was measured as 0.063% at 450 nm by taking TEOA as the sacrificial reagent (see details in Supplementary Fig. 29). By optimizing the Pt content and the type of sacrificial reagent, the hydrogen evolution amount of ZnPor-DETH-COF could be further improved (Supplementary Fig. 30), and AQE was determined to be 0.32%.

From TEM characterization, Pt nanoparticles (NPs) with similar morphology that identified by the characteristic crystal spacing (2.2 Å) for (111) lattice plane were formed in situ on four COFs with an average size of 3.5 nm under light irradiation (Supplementary Figs. 32 and 33), further confirming the varied HER activity was a result of the different photosensitization effect of the four COFs. From PXRD experiments, the crystal structures of all four COFs were well-retained after photocatalytic HER (Supplementary Figs. 34–37). In addition, these COFs showed excellent durability of photocatalytic hydrogen evolution. For example, the hydrogen production rate of ZnPor-DETH-COF is preserved even after 120 h irradiation (Fig. 4b).

In order to confirm the durability of electronic and chemical structures for the metal ion centers in photocatalysis process, we then performed X-ray absorption spectroscopy (XAS) measurement of three metalloporphyrin-based COFs (CoPor-DETH-COF, NiPor-DETH-COF, and ZnPor-DETH-COF) before and after hydrogen evolution. In the E-space curves, the M K-edge absorption of these three COFs remained unchanged during the reaction (Supplementary Fig. 38), confirming the invariability of the electronic configuration of metal centers. From Fourier transform R-space spectra (Fig. 5a–c), these three COFs still showed two main coordinated peaks after photocatalysis, which could be attributed to the unaltered primary M–N bond and secondary M–C coordination layer. In addition, their coordination numbers (CNs) of M–N and M–C were determined as ~4 and ~8 (see fitting details in Supplementary Table 7), strongly indicating the retained metalloporphyrin structures. Moreover, continuous Cauchy wavelet transform (CCWT) analysis displayed two intensity maximums, which were totally different from the standard Co, Ni, and Zn foils (Fig. 5d–l). Based on XAS R-space and CCWT analysis, we believe the single-atom characteristic of metal ion centers in all of these three COFs are well retained, which can clearly rule out the formation of metal clusters that might serve as catalytic center in the process of hydrogen photogeneration.

**Kinetic analysis of photocatalytic processes.** As the hydrogen evolution was performed in alkaline condition, the reductive half-reaction occurs via following reaction: $2H_2O + 2e^- \rightarrow 2OH^- + H_2$. We calculated the interaction between $H_2O$ and metalloporphyrin-based COFs (CoPor-DETH-COF, NiPor-DETH-COF, and ZnPor-DETH-COF) by density functional theory (DFT) (see models in Supplementary Fig. 41). Accordingly, the $H_2O$ absorption energy change ($\Delta E$) at CoPor-DETH-COF, NiPor-DETH-COF, and ZnPor-DETH-COF is 0.554, 0.847, and 0.659 eV, respectively (Supplementary Fig. 42). The enormously positive values strongly indicate the adsorption and activation of $H_2O$ molecules at the metal centers is unfavorable[56], which matches well with the negligible hydrogen production activity in the absence of Pt-cocatalysts (Supplementary Fig. 27). Therefore, it can be concluded that these COFs are light absorbers and can be photoexcited to produce electron-hole pairs, and the electrons migrate to the photo-deposited Pt NPs for hydrogen evolution while holes are consumed by sacrificial reagents. The concrete changes in the excited state and the whole catalytic processes can be found in Supplementary Figs. 39 and 40.

Then, the different photocalytical $H_2$ generation efficiency of MPor-DETH-COFs was studied by DFT calculations. As shown in Fig. 6 and Supplementary Figs. 43 and 44, the electron density is trapped within the porphyrin cores, thus lowering the in-plane charge transport. In consideration of their AA stacking structures, the out-of-plane charge carrier migration of these four COFs might become the main pathway. To validate this point, the projected density of states (PDOS) for their monolayer and bilayer structures are calculated (Supplementary Fig. 45). Compared with monolayer structures, the band gaps of bilayer counterparties are narrowed ~0.2–0.4 eV due to the interlayer π–π interaction (Supplementary Table 8), indicating the favored interlayer charge carrier migration[57].

In principle, considering the AA stacking structures of MPor-DETH-COFs, the macrocycle-on-macrocycle and metal-on-metal channels within porphyrin columns would play vital roles in the kinetics of charge-carrier separation and migration[21]. Usually, upon light excitation of the porphyrin π-ring, the photogenerated electron migration relies on metal-on-metal channel [$M_n^{2+} + e^- \rightarrow M_n^{(2n-1)+}$ ($n \gg 1$, M represents the metal ion)] rather than localized on a specific metal center, while photogenerated holes mainly transfer through macrocycle-on-macrocycle pathway. For $H_2$Por-DETH-COF, as no metal exists, both electron and hole migration proceed via macrocycle-on-macrocycle pathway (Fig. 6a), which will increase the possibility of charge recombination and thus lead to a short emission lifetime. However, for CoPor-DETH-COF and NiPor-DETH-COF, ligand-to-metal charge transfer (LMCT) process can be taken into

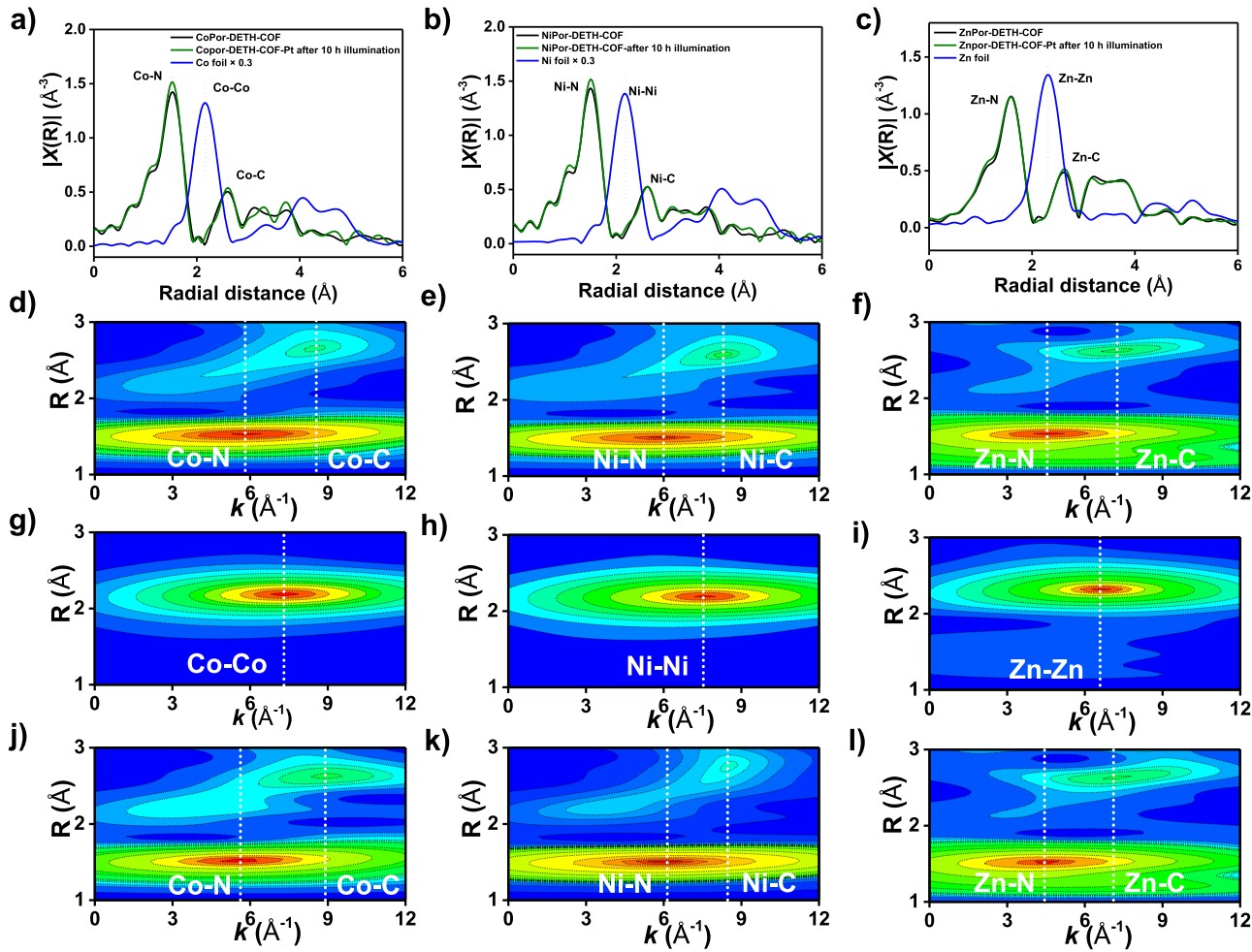

**Fig. 5 XAS analysis.** XAS Fourier transform R-space curves of **a** CoPor-DETH-COF, **b** NiPor-DETH-COF and **c** ZnPor-DETH-COF before and after light reaction; CCWT plots of **d** CoPor-DETH-COF, **e** NiPor-DETH-COF, and **f** ZnPor-DETH-COF before reaction; CCWT plots of corresponded metal foil: **g** Co, **h** Ni, and **i** Zn; CCWT plots of corresponded COFs after light reaction: **j** CoPor-DETH-COF, **k** NiPor-DETH-COF, and **l** ZnPor-DETH-COF.

consideration, since it significantly restrains the hole migration via macrocycle-on-macrocycle channel. Specifically, for CoPor-DETH-COF, LMCT process is preeminent owing to the $3d^7$ configuration of $Co^{2+}$, which suppresses holes migration (Fig. 6 and Supplementary Figs. 43 and 44). As a result, CoPor-DETH-COF showed the worst activity of hydrogen evolution. With the increase of $d$-electrons ($3d^8$ for $Ni^{2+}$), the LMCT process is partially suppressed, and hole transfer ability through macrocycle-on-macrocycle channel will be improved. Finally, in the case of $Zn^{2+}$ ion with $3d^{10}$ configuration, the LMCT process is strictly forbidden (the variation of center metal electrons density from $Co^{2+}$ to $Zn^{2+}$ can be clearly seen in Fig. 6). Therefore, the holes of ZnPor-DETH-COF can freely migrate via macrocycle-on-macrocycle channel to the surface and the electrons transfer via $Zn\cdots Zn$ chain, which will result in the long-time charge-separation state. Accordingly, ZnPor-DETH-COF demonstrates the highest activity toward photocatalytic hydrogen evolution under the identical conditions.

## Discussion

In summary, in order to explore the structure–property–activity relationship in photocatalytic HER from a molecular level, we have reported the designed synthesis and characterization of four isostructural porphyrinic 2D COFs, which have high crystallinity and large pore surface. Interestingly, by incorporating different transition metals into the porphyrin rings, the photophysical and

electronic properties of the porphyrinic COFs are adjusted. More importantly, these COFs showed tunable photocatalytic hydrogen production rate, mainly ascribed to their tailored charge-carrier dynamics via molecular engineering. Consequently, we believe the charge-carrier dynamics of COFs play a very important role in the photocatalytic HER from water. This study not only represents a simple example to efficiently tune the photocatalytic hydrogen evolution activities of COFs at molecular level, but also provides valuable insight on the structure design COFs for better photocatalytic performance in future. The construction of efficient COF-based photocatalysts (e.g., $CO_2$ reduction[46]) is undergoing in the lab.

## Methods

**Synthesis of $H_2$Por-DETH-COF.** A pyrex tube was charged with $p$-Por-CHO (16 mg, 0.022 mmol), DETH (13.04 mg, 0.044 mmol), 1,2-dichlorobenzene (0.5 mL), butanol (0.5 mL) and aqueous acetic acid (0.5 mL, 12 M). After being degassed through three freeze-pump-thaw cycles and then sealed under vacuum, the tube was heated at 120 °C for 3 days. The resulting precipitate was collected by centrifugation, exhaustively washed by Soxhlet extractions with THF and DCM for 24 h, dried under vacuum at 80 °C. The $H_2$Por-DETH-COF was isolated as brown powders in 88% yield. Elemental analysis: calculated C (70.92%), H (4.79%), N (13.78%) and observed C (69.14%), H (4.73%), N (13.24%).

**Synthesis of CoPor-DETH-COF.** A pyrex tube was charged with $p$-CoPor-CHO (17.23 mg, 0.022 mmol), DETH (13.04 mg, 0.044 mmol), 1,2-dichlorobenzene (0.8 mL), butanol (0.2 mL) and aqueous acetic acid (0.1 mL, 3 M). After being

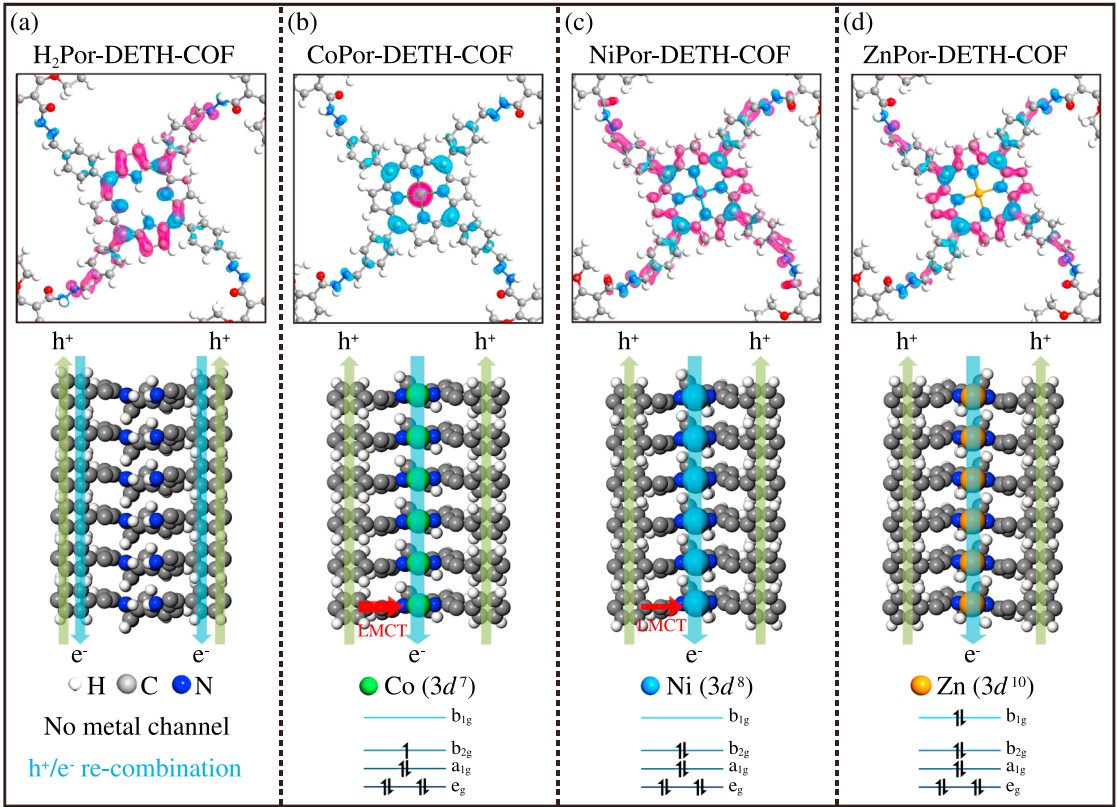

**Fig. 6 LMCT mechanism indicated by DFT calculations.** The isosurface of the electron orbitals of blue VBM and magenta CBM (upper panel) and schematic illustrations of the hole-electron transport processes (lower panel) in MPor-DETH-COFs: **a** H₂Por-DETH-COF; **b** CoPor-DETH-COF; **c** NiPor-DETH-COF, and **d** ZnPor-DETH-COF. The balls in different colors represent different atoms: H, white; C, grey; N, blue; Co, olive green; Ni, light blue; Zn, orange.

degassed through three freeze-pump-thaw cycles and then sealed under vacuum, the tube was heated at 120 °C for 7 days. The resulting precipitate was collected by centrifugation, exhaustively washed by Soxhlet extractions with THF and DCM for 24 h, dried under vacuum at 80 °C. The CoPor-DETH-COF was isolated as dark red powders in 85% yield. Elemental analysis: calculated C (67.76%), H (4.42%), N (13.17%) and observed C (66.69%), H (4.30%), N (12.37%).

**Synthesis of NiPor-DETH-COF.** A pyrex tube was charged with *p*-NiPor-CHO (17.22 mg, 0.022 mmol), DETH (13.04 mg, 0.044 mmol), 1,2-dichlorobenzene (0.5 mL), butanol (0.5 mL) and aqueous acetic acid (0.1 mL, 6 M). After being degassed through three freeze-pump-thaw cycles and then sealed under vacuum, the tube was heated at 120 °C for 7 days. The resulting precipitate was collected by centrifugation, exhaustively washed by Soxhlet extractions with THF and DCM for 24 h, dried under vacuum at 80 °C. The NiPor-DETH-COF was isolated as red powders in 86% yield. Elemental analysis: calculated C (67.77%), H (4.42%), N (13.17%) and observed C (66.02%), H (4.41%), N (12.62%).

**Synthesis of ZnPor-DETH-COF.** A pyrex tube was charged with *p*-ZnPor-CHO (17.38 mg, 0.022 mmol), DETH (13.04 mg, 0.044 mmol), 1,2-dichlorobenzene (0.8 mL), butanol (0.2 mL), and aqueous acetic acid (0.5 mL, 12 M). After being degassed through three freeze-pump-thaw cycles and then sealed under vacuum, the tube was heated at 120 °C for 7 days. The resulting precipitate was collected by centrifugation, exhaustively washed by Soxhlet extractions with THF and DCM for 24 h, dried under vacuum at 80 °C. The ZnPor-DETH-COF was isolated as green powders in 85% yield. Elemental analysis: calculated C (67.42%), H (4.40%), N (13.10%) and observed C (64.96%), H (4.38%), N (12.16%).

**Photocatalysis experiment.** The H₂ photogeneration test was performed with a 20 mL pyrex tube holding MPor-DETH-COF (2.5 mg), 5 mL phosphate buffer solution (0.1 M, pH = 7.0). The suspension was ultrasonicated for 30 min before adding 2.5 μL 8 wt% H₂PtCl₆ and 50 μL triethanolamine (TEOA), and then degassing by Ar bubbling for 30 min. Six hundred microliter of CH₄ was injected into the system and functioned as the internal standard for quantitative analysis. Xe lamps (300 W) as light source for testing H₂ evolution performance, and using air fan to keep room temperature of the sample. The generated H₂ gas in the headspace of reactor was taken with a gas-tight syringe and measured by using a gas chromatograph (Shimadzu GC2014CAFC/APC) equipped with a thermal conductivity detector and a 5 Å molecular sieves GC column. Ar was used as a carrier gas.

## Data availability

The data that support the plots within this paper and other findings of this study are available from the corresponding author on request.

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

## Acknowledgements

C.W. gratefully acknowledges financial support from the National Natural Science Foundation of China (21572170, 21772149 and 21975188) and the Fundamental Research Funds for Central Universities (2042020kf0213). L.-Z.W. and X.-B.L. are grateful for the financial support from the National Natural Science Foundation of China (21971251) and the National Key Research and Development Program of China (2017YFA0206903). L.G. and L.J.Q. thank financial support from the National Key Research and Development Program of China (2017YFB0702100). We also thank the Beijing Synchrotron Radiation Facility (BSRF, Beamline 1W1B) for providing beam time for the XAS measurements.

## Author contributions

R.C. performed the synthesis and characterization of COFs, including NMR, PXRD, FT-IR and gas absorption. Y.W. carried out $H_2$ evolution test, XAS data analysis and charge dynamics investigation. Y.M. did the structure stimulations and DFT calculations. X.-Y.G. helped with the $H_2$ evolution test and data analysis. L.G. and L.J.Q. guided the relevant calculation work. C.W., X.-B.L., and L.-Z.W. designed and supervised the project. R.C., Y.W., Y.M., A.M., X.-B.L., L.-Z.W., and C.W. analyzed the data and wrote the paper.

## Competing interests

The authors declare no competing interests.
