## [Peer Review File · Nature Communications]

REVIEWER COMMENTS

Reviewer #1 (Remarks to the Author):

In recent years, 2D covalent organic frameworks (COFs) have gained amount attention in photocatalysis, e.g. hydrogen evolution. However, the structure-property-activity relationship of photocatalytic COFs is still unclear and needs to be further explored. In this manuscript, Chen and coworkers report the synthesis of four isostructural porphyrinic 2D COFs (MPor-DETH-COF, M = H₂, Co, Ni, Zn) and their photocatalytic activity in hydrogen generation. The structures of these four COFs before and after photocatalysis have been well characterized by XRD, SEM, TEM, and XAS. More importantly, the incorporation of different transition metals into the porphyrin rings can rationally tune the photocatalytic hydrogen evolution rate of corresponding COFs. From a combination of time-resolved spectroscopic experiments and DFT calculations, this tunable performance can be mainly explained by their tailored charge-carrier dynamics via molecular engineering. In my opinion, this is a very nice paper to systematically reveal the structure-property-activity relationship of covalent organic frameworks as the light absorbers for photocatalytic hydrogen evolution. I would like to recommend the publication in Nature Communications after addressing the following minor concerns:

Q1. The experimental details of the X-ray absorption spectroscopy experiments should be described in the manuscript. Please include details of measurement conditions (i.e., solid state or liquid state), sample preparation, and data collection mode, etc.

Q2. In photocatalytic experiments, TEOA was employed as the sacrificial reagent. How about some other reagents (e.g., ascorbic acid, TEA, or Na₂SO₃) were used in the system?

Q3. In my understanding, the number of active sites is very important for the performance of catalysis, either electrocatalysis or photocatalysis. The author should clearly claim the influence of cocatalyst amount in the hydrogen evolution performance of these COFs.

Q4. In SI, the synthesis of H₂Por-DETH-COF should be Supplementary Scheme 4, but not 1. The Supplementary Figure 31 is not very clear. Please carefully revise the main text and SI.

Reviewer #2 (Remarks to the Author):

This paper includes lots of characterization data and asks an interesting question: Can the dependence of hydrogen evolution on porphyrin COF composition be understood? Unfortunately, the paper has, at least in its present form, some serious flaws that, in my view, make it unpublishable. Some of the issues:

1. Emission spectra need to be reported. It's very hard to believe that emission can be observed in the vicinity of 470 to 500 nm when electronic absorption extends to ~700 nm or beyond. I don't see how TRSPC signals at 470 to 500 nm can be reporting on the excited-state lifetimes of porphyrin COFs.

2. I may be wrong, but I don't think there's any literature evidence for luminescence from cobalt porphyrins. Why does a COF based on cobalt porphyrin units even emit light?

3. COFs based on zinc porphyrins are electronically very similar to COFs based on free-base porphyrins and the authors correctly argue that hole transport is macrocycle-based in both. But, they conclude that moving holes through the macrocycle facilitates charge-recombination for the free-base version yet facilitates charge-separation for the zinc version.

4. For the series of COFs, the weighted lifetimes derived from luminescence transients differ by only about a factor of three. The cobalt porphyrin COF, if it is anything at all like the isolated building block, should be essentially non-emissive and have an excited-state lifetime of perhaps 10 ps, i.e. hundreds of times shorter than the few nanosecond lifetime claimed here.

5. For the free-base and zinc-porphyrin based COFs, I'd expect the lowest singlet excited state to decay mainly via intersystem crossing to a much longer-lived triplet excited state, with only small percentage of the singlet decay occurring by fluorescence. Can triplet excited-state photochemistry be ruled out here?

6. As the authors point out, the COFs are light-absorbers/sensitizers and the actual catalysts are

platinum nanoparticles obtained by photochemical reduction of solution-phase Pt(2+) in the presence of a sacrificial electron donor, TEOA. The TOC cartoon shows ~1 nm Pt nanoparticles residing in COF ~3 nm COF channels. What do we really know about the loading, siting, and size dispersity of the actual catalyst, i.e. nanoparticulate Pt?

7. While not central to the paper, the authors should read about the complications of sacrificial reagents such as TEOA. After donating one electron, this particular sacrificial reagent decomposes in a way that yields a strongly reducing product that itself is capable of donating an electron. Typically, capture of one hole by TEOA yields two electrons – meaning that only one photon is needed to generate the two electrons required to form H₂ from water via platinum as a catalyst.

8. How are excited-state lifetimes changed by exposure to TEOA and platinum? Given a ca. 40-fold difference in H₂ production rates across the series of COFs, and given the effectiveness of Pt as a hydrogen evolution catalyst, I'd expect the lifetime of the least effective photosensitizer, free-base porphyrin COF to be affected hardly at all and the excited-state lifetime of the most effective photosensitizer to be greatly decreased.

Reviewer #3 (Remarks to the Author):

The manuscript by Chen and co-workers and the work discussed therein is interesting, especially the role of the metal cations in the electron transport, but I have some reservations about how the work was performed and reported:

- On page 9 the authors write "The amount of hydrogen evolved from the most active ZnPor-DETH-COF is comparable with those reported COFs-based photocatalytical systems, such as TP-BDDA37 and g-C18N3-COF39." Now comparing rates with literature data measured using different set-up is fraught with difficulties and I as a referee would not insist on it, but if the authors insist to do so they should probably also compare with Wang et al. (Nat. Chem. 10, 1180–1189 (2018), ref. 42), who report rates twice of the best material here.
- I personally am no big fan of the way the authors show the hydrogen evolution data vs. time in Fig. 3A. They should at least include the same data in the form of one conventional scatter plot per material in the supporting material.
- There appears to be no quantum efficiency data. Something I would have expected to be there, at least for the best performing material.
- Looking at Fig. S1 I'm rather surprised with the confidence that the authors quote HOMO and LUMO values in the main text. These CVs, as is common for solids, are so far away from what the textbook CV looks like that I personally wouldn't be confident to associate any particular feature to an oxidation or reduction process.
- The predicted adsorption enthalpies are surprisingly strongly positive. Cobalt(II), for example, in solution forms octahedral complexes with water, yet here the authors predict that adding a water molecule to tetrahedrally coordinated cobalt is endothermic.
- The authors might want to explain more clearly how LMCT restrains hole transfer. This to me does not feel trivial. They also might want to explain more clearly what Figure 5 and Supplementary Figure 31 show regarding LMCT processes.
- The authors should probably make all relevant DFT/DFTB optimised structure available in the form of machine readable supporting information.

Point-By-Point Response to the Reviewers' Comments

Reviewer #1:

Comment 1: *In recent years, 2D covalent organic frameworks (COFs) have gained amount attention in photocatalysis, e.g. hydrogen evolution. However, the structure-property-activity relationship of photocatalytic COFs is still unclear and needs to be further explored. In this manuscript, Chen and coworkers report the synthesis of four isostructural porphyrinic 2D COFs (MPor-DETH-COF, M = H₂, Co, Ni, Zn) and their photocatalytic activity in hydrogen generation. The structures of these four COFs before and after photocatalysis have been well characterized by XRD, SEM, TEM, and XAS. More importantly, the incorporation of different transition metals into the porphyrin rings can rationally tune the photocatalytic hydrogen evolution rate of corresponding COFs. From a combination of time-resolved spectroscopic experiments and DFT calculations, this tunable performance can be mainly explained by their tailored charge-carrier dynamics via molecular engineering. In my opinion, this is a very nice paper to systematically reveal the structure-property-activity relationship of covalent organic frameworks as the light absorbers for photocatalytic hydrogen evolution. I would like to recommend the publication in Nature Communications after addressing the following minor concerns:*

Response: We thank the reviewer for these very positive comments and support the publication of our work. We have carefully modified the manuscript and the corresponding changes have been added in the revised manuscript.

Comment 2: *The experimental details of the X-ray absorption spectroscopy experiments should be described in the manuscript. Please include details of measurement conditions (i.e., solid state or liquid state), sample preparation, and data collection mode, etc.*

Response: We thank the reviewer for bringing this point to our attention. According to this valuable suggestion, we have added experimental details for the XAS measurements in the revised SI on page S4, as followed:

X-ray Absorption spectroscopy (XAS) was acquired at beamline 1W1B at the Beijing Synchrotron Radiation Facility (BSRF) with Si (111) double-crystal monochromator. Under the condition of dedicated synchrotron light, the ring energy is 2.5 GeV and the total beam current is 250 mA in top-up mode. Before XAS data collection, K-edge energy calibration was performed with corresponding metallic foil standards (Co, Ni, and Zn), and the data collection was carried out in transmission mode using ionization chamber. For MPor-DETH-COFs (M = Co, Ni, Zn), solid state samples were evenly milled and smeared onto a metal-free polyimide tape, and then the data was collected in fluorescence mode using a Lytle detector.

Extended X-ray absorption fine structure (EXAFS) spectra were transformed into *R*-space by Athena software. Firstly, the XAS spectra were obtained by subtracting the pre-edge background (−150 to −50 eV vs. absorption edge) from the overall

absorption and then normalized with range of 150–700 eV. Subsequently, $\chi(k)$ data in the k -space were Fourier transformed to R -space using a hanning window (k -weight = 2, k is ranged from 3.0 to 12.0 \AA^{-1}) to separate the EXAFS contributions from different coordination shells. EXAFS spectra were fitted by Artemis software and amplitude attenuation factor (amp) was calculated from corresponding metal foil. M–N and M–C single scattering paths were extracted from metal porphyrin model. The continuous Cauchy wavelet transform (CCWT) was carried out by using Larch 0.9.35 software (k weight = 2).

Comment 3: In photocatalytic experiments, TEOA was employed as the sacrificial reagent. How about some other reagents (e.g., ascorbic acid, TEA, or Na_2SO_3) were used in the system?

Response: We thank the reviewer for this valuable comment. As per the reviewer's suggestion, we investigated the photocatalytic performance in the presence of different sacrificial reagents ($\text{Na}_2\text{S}/\text{Na}_2\text{SO}_3$, TEA, TEOA and ascorbic acid), by taking ZnPor-DETH-COFs as an example. As shown in Figure R1, these sacrificial reagents exhibit different hydrogen evolution activity under the same condition, and among them, ascorbic acid shows the highest value. The corresponding results have been added in the revised SI as Supplementary Figure 26b.

Figure R1. Photocatalytic H_2 evolution of ZnPor-DETH-COF (2.5 mg in 5 mL H_2O) in the presence of different sacrificial reagents [Na_2SO_3 : 0.126 mmol, Na_2S : 0.126 mmol; TEOA: 50 μL ; TEA: 50 μL ; Ascorbic acid: 0.126 mmol (pH~8.0), reaction time: 2 h; light source: 450 nm LED lamps].

Comment 4: In my understanding, the number of active sites is very important for the performance of catalysis, either electrocatalysis or photocatalysis. The author should clearly claim the influence of cocatalyst amount in the hydrogen evolution performance of these COFs.

Response: We thank the reviewer for this valuable comment. Indeed, the number of active sites plays a very important role in the photocatalytic performance. We investigated the photocatalytic H_2 evolution of ZnPor-DETH-COF in the presence of different amounts of cocatalyst precursor (H_2PtCl_6). As shown in Figure R2, the rates

of H₂ production in 2 h enhanced with the increase of the concentration of Pt precursor and reached a maximum when 10 μL H₂PtCl₆ (8 wt%) were introduced. The corresponding changes have been added in the revised manuscript and SI as Supplementary Figure 26a.

Figure R2. H₂ photoproduction versus different volume of H₂PtCl₆ (8 wt%, μL).

Comment 5: In SI, the synthesis of H₂Por-DETH-COF should be Supplementary Scheme 4, but not 1. The Supplementary Figure 31 is not very clear. Please carefully revise the main text and SI.

Response: We thank the reviewer for bringing this point to our attention and pointing out our careless. We have checked all the Figure numbers again and also replaced Supplementary Figure 39 with a much higher quality picture.

Reviewer #2:

Comment 1: *This paper includes lots of characterization data and asks an interesting question: Can the dependence of hydrogen evolution on porphyrin COF composition be understood? Unfortunately, the paper has, at least in its present form, some serious flaws that, in my view, make it unpublishable. Some of the issues:*

Response: We thank the reviewer for interesting in our work. As this reviewer may agree, since Lotsch and co-workers reported the first example in 2014, the utilizing of 2D COF in photocatalytic hydrogen evolution has gained increasing attention, but the rational tuning of their structures and photophysical properties for maximizing the hydrogen evolution efficiency needs to be further clarified. Therefore, we tried to establish the relationship between hydrogen evolution activity and COF composition from a molecular level. In this paper, we designed four isostructural 2D COFs with tunable optoelectronic properties and investigated their photocatalytic activity in hydrogen generation. Based on our results, we believe the charge-carrier dynamics of COFs play a very important role in the photocatalytic HER from water. We believe this study not only represents a simple and effective way for efficient tuning of the photocatalytic hydrogen evolution activities of COFs at molecular level, but also provides valuable insight on the structure design for better COFs photocatalysis in future.

According to the reviewer's valuable suggestions, we have performed related spectroscopic data again and carefully analyzed the results in the revised manuscript. Indeed, we found some characterization is wrong and CoPor-DETH-COF is non-emissive (we thank this reviewer very much for pointing out our mistake). However, we should claim here, our conclusion is definitely right. The corresponding changes have been added into the revised manuscript.

Comment 2: *Emission spectra need to be reported. It's very hard to believe that emission can be observed in the vicinity of 470 to 500 nm when electronic absorption extends to ~700 nm or beyond. I don't see how TRSPC signals at 470 to 500 nm can be reporting on the excited-state lifetimes of porphyrin COFs.*

Response: We thank the reviewer for these professional comments and pointing out our mistake. According to literature reports (*Angew. Chem. Int. Ed.* **2012**, *51*, 7440; *J. Photoch. Photobio. A* **2019**, *375*, 91; *J. Am. Chem. Soc.* **2020**, *142*, 705; *Spectrochim. Acta. A* **2020**, *240*, 118570), the emission peak of porphyrin and its derivatives is mainly distributed in the range of 550–800 nm. In our former manuscript, we mistakenly attributed the Raman scattering peak in the vicinity of 470 to 500 nm to the luminescence peak. Therefore, in the revised manuscript, we performed the relevant spectral data again by adding a 550 nm cut-off filter. As shown in Figure R3a, the mainly emission peaks of MPor-DETH-COFs were observed in the range of 650–750 nm. Compared with H₂Por-DETH-COF, NiPor-DETH-COF and ZnPor-DETH-COF, CoPor-DETH-COF is non-emissive. Consequently, at the position of highest emission point, we re-measured the TRSPC signals of MPor-DETH-COFs (M = H₂, Ni and Zn). As can be seen in Figure R3b, the amplitude-weighted average

lifetimes follow the order of ZnPor-DETH-COF > NiPor-DETH-COF > H₂Por-DETH-COF, which is consistent with the performance of H₂ photogeneration. The corresponding changes have been added into the revised manuscript on page 8 and Figure 2c.

Figure R3. (a) PL spectra of MPor-DETH-COFs (Excitation: 405 nm). (b) The emission decay of MPor-DETH-COFs (Excitation: 405 nm; signal position: 682 nm for H₂Por-DETH-COF and ZnPor-DETH-COF, and 660 nm for NiPor-DETH-COF).

Comment 3: *I may be wrong, by I don't think there's any literature evidence for luminescence from cobalt porphyrins. Why does a COF based on cobalt porphyrin units even emit light?*

Response: We thank the reviewer for this valuable comment and pointing out our mistake. According to the literatures, cobalt (III) porphyrins (*J. Am. Chem. Soc.* **2019**, *141*, 9155) or cobalt (II) porphyrin with axial ligands (*Inorg. Chem. Commun.* **2020**, *118*, 107995) and asymmetric structures (*ACS Appl. Energy Mater.* **2019**, *2*, 5665) can exhibit luminescence. For symmetrical cobalt (II) porphyrins in the absence of axial ligands, there is no evidence to show obvious luminescence. In our case, porphyrin is coordinated with cobalt (II) and without axial ligands or symmetrical structures. Therefore, this COF should have no emission, which is also confirmed from the revised emission spectrum (Figure R3a). Accordingly, we have removed the TRSPC signal of CoPor-DETH-COF in the revised manuscript.

Comment 4: *COFs based on zinc porphyrins are electronically very similar to COFs based on free-base porphyrins and the authors correctly argue that hole transport is macrocycle-based in both. But, they conclude that moving holes through the macrocycle facilitates charge-recombination for the free-base version yet facilitates charge-separation for the zinc version.*

Response: We thank the reviewer for this comment. From the perspective of LMCT processes, ZnPor-DETH-COF are electronically very similar to the counterparts based on H₂Por-DETH-COF, according to their isosurface of the electronic orbitals (Figure R4 top). According to the literature (*Angew. Chem. Int. Ed.* **2012**, *51*, 2618), the metal center can act as an electron channel to facilitate the migration of electrons in ZnPor-DETH-COF, and the macrocycles have contributed to hole transport. However,

in the H₂Por-DETH-COF, both electrons and holes migrate through the macrocycle channels, and the photogenerated charges might re-combine due to the lack of efficient spatial separation (Figure R4 bottom).

Figure R4. The isosurface of the electronic orbitals of valence band maximums (VBM, blue) and conduction band minimums (CBM, magenta) and schematic illustration of the hole-electron transport processes in H₂Por-DETH-COF (a) and ZnPor-DETH-COF (b).

Comment 5: For the series of COFs, the weighted lifetimes derived from luminescence transients differ by only about a factor of three. The cobalt porphyrin COF, if it is anything at all like the isolated building block, should be essentially non-emissive and have an excited-state lifetime of perhaps 10 ps, i.e. hundreds of times shorter than the few nanosecond lifetime claimed here.

Response: We thank the reviewer for this professional comment. Indeed, cobalt porphyrin is non-emissive as the excited state rapidly deactivates to a low-lying dd state, which decays in *ca.* 10 picoseconds (*J. Phys. Chem.* **1993**, 97, 8969). As shown in Figure R3a (*vide supra*), the CoPor-DETH-COF is non-emissive. Therefore, we have removed the TRSPC signal of CoPor-DETH-COF in the revised manuscript.

Comment 6: For the free-base and zinc-porphyrin based COFs, I'd expect the lowest singlet excited state to decay mainly via intersystem crossing to a much longer-lived triplet excited state, with only small percentage of the singlet decay occurring by fluorescence. Can triplet excited-state photochemistry be ruled out here?

Response: We thank the reviewer for bringing this point to our attention. The intersystem crossing is prone to occur for free-based porphyrin and Zn-porphyrin (please see the *Jablonski* diagram in Scheme R1a). However, we didn't observe an obvious phosphorescent signal from MPor-DETH-COFs (M = H₂ and Zn) under the vacuum condition. According to the perturbation theory and compared with the monomer, the energy gap (ΔE_{st}) between the lowest singlet and triplet excited state of the polymer will be greatly reduced (*J. Am. Chem. Soc.* **2019**, *141*, 5045; *Chem. Rev.* **1966**, *66*, 199; *Angew. Chem., Int. Ed.* **2018**, *57*, 6449), which is more conducive to the transition from $*T_n$ to $*S_1$ and returns to S₀ state to generate thermally activated delayed fluorescence (TADF). Consequently, MPor-DETH-COFs (M = H₂ and Zn) didn't show the phosphorescence and the rapid reverse intersystem crossing (RISC) process makes the charge separation process mainly occur in the excited singlet state $*S_1$.

In fact, as shown in Figure R5, the long-lived decay of free-base and zinc-porphyrin based COFs can be detected under the vacuum condition, which can be attributed to TADF process (*J. Mater. Chem. A* **2020**, *8*, 3005). The weights of TADF for free-base and zinc-porphyrin based COFs are calculated as 23.0% and 29.6%. Considering the rate of hole transfer is about two to four orders of magnitude slower than that of electrons (*Proc. Natl. Acad. Sci. USA* **2011**, *108*, 29; *Chem. Commun.* **2013**, *49*, 4400), we infer that before ISC process, the excited state COF preferentially undergoes an oxidative quenching process with Pt NPs to produce COF^{•+} (*Inorg. Chem.* **2020**, *59*, 1611). This fluorescence process (short-lived singlet) is an important indicator of the photocatalytic hydrogen evolution process. Then, COF^{•+} is reductively quenched by acquiring an electron from TEOA to return to the ground state (Scheme R2b). The corresponding changes have been added into the revised SI as Supplementary Figure 35.

Scheme R2. The *Jablonski* process (a) and charge separation diagram (b) of MPor-DETH-COFs.

Figure R5. Comparison of the emission decay of H₂Por-DETH-COF and ZnPor-DETH-COF (Excitation: 405 nm; signal position: 682 nm; room temperature under the condition of oil pump vacuum).

Comment 7: As the authors point out, the COFs are light-absorbers/sensitizers and the actual catalysts are platinum nanoparticles obtained by photochemical reduction of solution-phase Pt(2+) in the presence of a sacrificial electron donor, TEOA. The TOC cartoon shows ~1 nm Pt nanoparticles residing in COF ~3 nm COF channels. What do we really know about the loading, siting, and size dispersity of the actual catalyst, i.e. nanoparticulate Pt?

Response: We thank the reviewer for these valuable comments. In the photocatalytic hydrogen evolution, Pt nanoparticles were in situ generated from H₂PtCl₆ precursor by accepting photoelectrons from MPor-DETH-COFs. Therefore, the loading of Pt on MPor-DETH-COFs can be calculated to ~3.8 wt%, according to the amounts of added H₂PtCl₆ and MPor-DETH-COFs in the photosystem. After photocatalytic experiments, the *in situ* generated Pt nanoparticles will load on the surface of MPor-DETH-COFs, as shown in Figure R6. Then, we carefully characterized the size distribution of Pt nanoparticles. As shown in Figure R7, the average diameter of Pt nanoparticles *in situ* generated after light deposition is determined to be about 3.5 nm, which is larger than the pore size of COF channels (~2.4 nm from DFT calculations). Therefore, the previous TOC is not reasonable and we changed the TOC cartoon graph (Figure R8). These changes have been added into the revised manuscript.

Figure R6. TEM images of a) H₂Por-DETH-COF, b) CoPor-DETH-COF, c) NiPor-DETH-COF, and d) ZnPor-DETH-COF after 10 h photocatalysis experiment.

Figure R7. Diameter distribution of Pt nanoparticles loaded on MPor-DETH-COF (M= H₂, Co, Ni, and Zn; statistic: 50).

Figure R8. Table of Content

Comment 8: While not central to the paper, the authors should read about the complications of sacrificial reagents such as TEOA. After donating one electron, this particular sacrificial reagent decomposes in a way that yields a strongly reducing product that itself is capable of donating an electron. Typically, capture of one hole by TEOA yields two electrons – meaning that only one photon is needed to generate the two electrons required to form H_2 from water via platinum as a catalyst.

Response: We thank the reviewer for these valuable comments. Indeed, as illustrated in Scheme R2, the TEOA could generate two electrons and two protons. After MPor-DETH-COFs absorbed one photon, they reached to an excited state, leading to charges separation. Since the rate of hole transfer is about two to four orders of magnitude slower than that of electrons (*Proc. Natl. Acad. Sci. USA* **2011**, 108, 29; *Chem. Commun.* **2013**, 49, 4400), the excited state COF will give priority to oxidation quenching that the electrons will be transferred from COF to Pt and produce $COF^{+\bullet}$. By acquiring an electron from TEOA, COF returns to the ground state, thus completing the catalytic cycle process. Subsequently, TEOA that has lost electrons ($TEOA_{ox}^{+\bullet}$) would extract H^+ from adjacent TEOA and produce $TEOA_{red}^{\bullet}$ (*J. Phys. Chem.* **1991**, 95, 7717). According to the literatures (*J. Am. Chem. Soc.* **1981**, 103, 369; *J. Am. Chem. Soc.* **2012**, 134, 11701; *Acc. Chem. Res.* **2015**, 48, 851; *Green Chem.* **2014**, 16, 1082), we speculated that, in our system, $TEOA_{red}^{\bullet}$ may spontaneously lose one electron to Pt NPs and form $(HOCH_2CH_2)_2N^+=CHCH_2OH$ itself, which is eventually degraded into $(HOCH_2CH_2)_2NH$ and $HOCH_2CHO$ by reacting with H_2O . As a result, the above processes can be written as: $(HOCH_2CH_2)_3N + H_2O \rightarrow (HOCH_2CH_2)_2NH + HOCH_2CHO + 2H^+ + 2e^-$, under the condition that COF just gain one photon. The detailed oxidation process of the TEOA was added in the revised SI as Supplementary Figure 36.

Scheme R2. Schematic illustration of TEOA oxidation processes.

Comment 9: How are excited-state lifetimes changed by exposure to TEOA and platinum? Given a ca. 40-fold difference in H_2 production rates across the series of

COFs, and given the effectiveness of Pt as a hydrogen evolution catalyst, I'd expect the lifetime of the least effective photosensitizer, free-base porphyrin COF to be affected hardly at all and the excited-state lifetime of the most effective photosensitizer to be greatly decreased.

Response: We thank the reviewer for these valuable comments! According to the reviewer's suggestion, we performed oxidation quenching and reduction quenching experiments, as shown in the Figure R9. We have successfully observed the influence of Pt content on the excited state lifetime of ZnPor-DETH-COF and H₂Por-DETH-COF in the short lifetime scale. For H₂Por-DETH-COF, after adding 10 μ L H₂PtCl₆, the lifetime is shortened from 0.82 ns to 0.50 ns for H₂Por-DETH-COF and the lifetime is changed from 1.40 ns to 0.68 ns for ZnPor-COF.

Since the rate of hole transfer is about two to four orders of magnitude slower than that of electrons (*Proc. Natl. Acad. Sci. USA* **2011**, 108, 29; *Chem. Commun.* **2013**, 49, 4400), the time scale between short-lived fluorescence decay and the hole migration process of COF might be mismatched, we cannot observe the quenching effect of TEOA on the COF excited state on a short lifetime scale (fluorescence decay). Taking into account the TADF process of H₂Por-DETH-COF and ZnPor-DETH-COF (*vide supra*, Figure R5), we observed the quenching effect of TEOA on ZnPor-DETH-COF and H₂Por-DETH-COF on a long-life time scale (vacuum condition). After adding 100 μ L 1% TEOA aqueous solution, the TADF lifetimes of H₂Por-DETH-COF and ZnPor-DETH-COF decreased from 498.30 and 558.66 ns to 138.57 and 75.05 ns, respectively. Therefore, it can be concluded that the charge separation ability of ZnPor-DETH-COF is significantly better than that of H₂Por-DETH-COF.

Figure R9. The emission decay of H₂Por-DETH-COF and ZnPor-DETH-COF before and after adding H₂PtCl₆ and TEOA solution: (a) H₂Por-DETH-COF (2.5 mg) and (b) ZnPor-DETH-COF (2.5 mg) with addition of different volume H₂PtCl₆ (8 wt% solution); (c) H₂Por-DETH-COF (2.5 mg) and (d) ZnPor-DETH-COF (2.5 mg) with addition of 100 μ L TEOA (1% aqueous solution).

Reviewer #3:

Comment 1: The manuscript by Chen and co-workers and the work discussed therein is interesting, especially the role of the metal cations in the electron transport, but I have some reservations about how the work was performed and reported:

Response: We thank the reviewer's positive comments expressed for our interesting result. We have addressed these concerns according to the suggestion in the revised manuscript.

Comment 2: On page 9 the authors write "The amount of hydrogen evolved from the most active ZnPor-DETH-COF is comparable with those reported COFs-based photocatalytic systems, such as TP-BDDA³⁷ and g-C₁₈N₃-COF³⁹." Now comparing rates with literature data measured using different set-up is fraught with difficulties and I as a referee would not insist on it, but if the authors insist to do so they should probably also compare with Wang et al. (*Nat. Chem.* 10, 1180–1189 (2018), ref. 42), who report rates twice of the best material here.

Response: We thank the reviewer for bringing this point to our attention. We fully agree with this reviewer that it is not reasonable to directly compare the activities of H₂ evolution in different works, due to the different experimental conditions (e.g., sacrificial reagent, light source, cocatalysts amount, etc., Shown as Table R1) in each Lab. Therefore, we have deleted the sentence "The amount of hydrogen evolved from the most active ZnPor-DETH-COF is comparable with those reported COFs-based photocatalytic systems, such as TP-BDDA³⁷ and g-C₁₈N₃-COF³⁹" in the revised manuscript.

Table R1. Summary of COFs based photocatalytic H₂ evolution systems.

2D COFs	Light source	Sacrificial reagent	Pt loading	HER rate	Reference
TFPT-COF	300 W Xe lamp (cut off 420 nm)	10% V TEOA	4 mg COF catalyst, 2.4 μ l (8 wt% H ₂ PtCl ₆)	1.9 mmol h ⁻¹ g ⁻¹ (5 h)	Chem. Sci. 2014 , 5, 2789
N ₃ -COF	300 W Xe lamp (cut off 420 nm)	1% V TEOA	5 mg COF catalyst, 5 μ l (8 wt% H ₂ PtCl ₆)	1.7 mmol h ⁻¹ g ⁻¹ (5 h)	Nat. Commun. 2015 , 6, 8508
TP-BDDA	300 W Xe lamp (cut off 395 nm)	10% V TEOA	3 wt%	324 \pm 10 μ mol h ⁻¹ g ⁻¹ (10 h)	J. Am. Chem. Soc. 2018 , 140, 1423.
A-TEBPY-COF	300 W Xe lamp (cut off 420 nm)	10% V TEOA	10 mg COF catalyst, 6 μ l (8 wt% H ₂ PtCl ₆)	98 μ mol h ⁻¹ g ⁻¹ (22 h)	Adv. Energy Mater. 2018 , 8, 1703278

g-C ₁₈ N ₃ -COF	300 W Xe lamp (cut off 420 nm)	1 M ascorbic acid	3 wt%	14.6 μmol h ⁻¹ g ⁻¹ (16 h)	J. Am. Chem. Soc. 2019 , 141 , 14272.
FS-COF	300 W Xe lamp (cut off 420 nm)	0.1 M ascorbic acid	5 mg COF catalyst, 5 μl (8 wt% H ₂ PtCl ₆)	10.1 ± 0.3 mmol h ⁻¹ g ⁻¹ (5 h)	Nat. Chem. 2018 , 10 , 1180
sp ² c-COF	300 W Xe lamp (cut off 420 nm)	10% V TEOA	3 wt%	1360 μmol h ⁻¹ g ⁻¹ (5 h)	Chem 2019 , 5 , 1632
ZnPor-DETH-COF	300 W Xe lamp (cut off 400 nm)	1% V TEOA	2.5 mg COF catalyst, 2.5 μl (8 wt% H ₂ PtCl ₆)	413 μmol h ⁻¹ g ⁻¹ (10 h)	This work

Comment 3: I personally am no big fan of the way the authors show the hydrogen evolution data vs. time in Fig. 3A. They should at least include the same data in the form of one conventional scatter plot per material in the supporting material.

Response: We thank the reviewer for bringing this point to our attention. We have also made a Figure with hydrogen evolution data vs. time (Figure R10), which was modified in the revised SI as Supplementary Figure 24.

Figure R10. Time dependent H₂ photogeneration using visible light for H₂Por-DETH-COF, CoPor-DETH-COF, NiPor-DETH-COF and ZnPor-DETH-COF (2.5 mg catalyst in 5 mL phosphate buffer solution, 2.5 μL (8 wt% H₂PtCl₆), 50 μL TEOA, λ > 400 nm 300 W Xe lamp).

Comment 4: There appears to be no quantum efficiency data. Something I would have expected to be there, at least for the best performing material.

Response: We thank the reviewer for this valuable comment. Indeed, quantum efficiency is a vital indicator for H₂ photogeneration. In the revised manuscript, we measured the apparent quantum efficiency (AQE) of ZnPor-DETH-COF by using the

device in Figure R11. Accordingly, ZnPor-DETH-COF showed a value of 0.063% (10 μ L TEOA and 3.8 wt% Pt). The detailed experimental information have been added in the revised SI on page S35.

Figure R11. The device for quantum efficiency measurement: (a) physical photos; (b) detailed schematic; (c) the optical power measurement process of LED lights.

Comment 5: Looking at Fig. S1 I'm rather surprised with the confidence that the authors quote HOMO and LUMO values in the main text. These CVs, as is common for solids, are so far away from what the textbook CV looks like that I personally wouldn't be confident to associate any particular feature to an oxidation or reduction process.

Response: We thank the reviewer for this comment. In fact, according to the literatures (*Science*, **2017**, 357, 673; *Angew. Chem. Int. Ed.* **2019**, 58, 6430 and *J. Am. Chem. Soc.*, **2018**, 140, 4623), it is quite common to employ CVs to measure the HOMO and LUMO values of COFs. In addition, we have also used UV-vis absorption spectrum to characterize the band gap of the MPor-DETH-COFs, and the result matched well with the HOMO and LUMO values measured from the CVs. Therefore, we believe that the HOMO and LUMO values of COFs determined by CV measurements are credible.

Comment 6: The predicted adsorption enthalpies are surprisingly strongly positive. Cobalt(II), for example, in solution forms octahedral complexes with water, yet here the authors predict that adding a water molecule to tetrahedrally coordinated cobalt is endothermic.

Response: We thank the reviewer for bringing this point to our attention. Indeed, the isolated Co(II) ion forms octahedral complexes with water in solution. However, in the COF, Co and N atoms form strong chemical bonds, which will reduce the capability of Co to adsorb water molecule as presented by DFT calculations. The electronic interactions between Co and N atoms could make Co(II) hydrophobic, indicating that the bonded Co(II) behaves quite differently from the isolated Co(II).

Comment 7: The authors might want to explain more clearly how LMCT restrains hole transfer. This to me does not feel trivial. They also might want to explain more clearly what Figure 5 and Supplementary Figure 31 show regarding LMCT processes.

Response: We thank the reviewer for these valuable comments. According to the literature (*Angew. Chem. Int. Ed.* **2012**, *51*, 2618), the metal center can act as an electron channel to facilitate the migration of electrons, while macrocycles will contribute to hole transport. For H₂Por-DETH-COF, the photogenerated charges might re-combine due to lack of efficient spatial separation for electron-hole pairs (Figure R12a). For other three COFs (ZnPor-DETH-COF, NiPor-DETH-COF and CoPor-DETH-COF), ligand-to-metal charge transfer (LMCT) process should be taken into consideration, as it significantly restrains the hole migration via macrocycle-on-macrocycle channel. Specifically, for CoPor-DETH-COF, LMCT process is preminent owing to the 3d⁷ configuration of Co²⁺, which suppresses holes migration (Figure S12b). As a result, CoPor-DETH-COF showed the worst activity of hydrogen evolution. With the increase of d-electrons (3d⁸ for Ni²⁺), the LMCT process is suppressed, and hole transfer ability through macrocycle-on-macrocycle channel will be improved. Finally, in the case of Zn²⁺ ion with 3d¹⁰ configuration, the LMCT is strictly forbidden (the variation of center metal electrons density from Co²⁺ to Zn²⁺ can be clearly seen in Figure R12). Therefore, the holes of ZnPor-DETH-COF can freely migrate via macrocycle-on-macrocycle channel to the surface, which will result in the long-time charge-separation state. Accordingly, ZnPor-DETH-COF demonstrates the highest activity toward photocatalytic hydrogen evolution under the identical conditions. In order to make it more clear, we have modified the Figure 5 in the revised manuscript and also added some description on page 12.

Figure R12. Schematic illustration of the hole-electron transport processes in MPor-DETH-COFs: (a) H₂Por-DETH-COF; (b) CoPor-DETH-COF; (c) NiPor-DETH-COF; (d) ZnPor-DETH-COF.

Comment 8: *The authors should probably make all relevant DFT/DFTB optimised structure available in the form of machine readable supporting information.*

Response: We thank the reviewer for bringing this point to our attention. We have attached relevant structures to supplementary materials with machine readable cif form.

REVIEWER COMMENTS

Reviewer #1 (Remarks to the Author):

The authors have addressed my questions and concerns. And I think I am fine with the acceptance for the manuscript.

Reviewer #2 (Remarks to the Author):

I can see that the authors did a lot of work in revising their paper, including removing several results that proved to be artifacts. The idea that a d-10 Zn(II) center can function as an electron conduit is really far-fetched from a coordination-chemistry and metalloporphyrin chemistry perspective. Zn(II) is a classic electronically inert metal ion in porphyrin and similar species, as there simply are no d-orbitals left to accommodate additional electrons. I know of no electrochemical or photochemical work that shows that Zn(II) can be reduced. The porphyrin ring is reduced instead. I recognize that the following observation applies to the molecular limit, but Zn(I) is an exceedingly rare oxidation state for zinc. The observations about differences in H₂ yield for free-base versus Zn(II) porph. species are likely correct, but the explanation seems at odds with well known chemistry.

Reviewer #3 (Remarks to the Author):

Having considered the revised manuscript and the reply to referees, I only have some comments on the reply of the authors to my comments (numbering refers to that of my original comments):

Comment 4: It is great that the authors have now measured an AQE value and report it in the supplementary information but they really in my opinion should also refer to it in the main manuscript. A reader shouldn't have to dig for this sort of information in the supplementary information.

The AQE value measured is rather low (0.063%) and much lower than that of other COFs reported in the literature, e.g. TP-BDDA-COF (1.8%), FS-COF (3.2%) and N3-COF (0.44%), making it even more important for the authors to discuss this value.

Comment 5: Fair enough, I am aware that other groups have used similar "quality" CVs to extract HOMO and LUMO values. However, to me the mere fact that other people for whom cyclic voltammetry most likely is a black box technique have done the same is perhaps not as convincing as for the authors.

Comment 7: Thanks for the explanation. I feel, however, it is being wasted in the reply to referees. I would suggest that the authors copy (part of) their reply to my comment 7 in the supplementary information into the main text.

Point-By-Point Response to the Reviewers' Comments

Reviewer #1

Comment: *The authors have addressed my questions and concerns. And I think I am fine with the acceptance for the manuscript.*

Response: We thank the reviewer for this positive comment and support the publication of our manuscript!

Reviewer #2

Comment: *I can see that the authors did a lot of work in revising their paper, including removing several results that proved to be artifacts. The idea that a d-10 Zn(II) center can function as an electron conduit is really far-fetched from a coordination-chemistry and metalloporphyrin chemistry perspective. Zn(II) is a classic electronically inert metal ion in porphyrin and similar species, as their simply are no d-orbitals left to accommodate additional electrons. I know of no electrochemical or photochemical work that shows that Zn(II) can be reduced. The porphyrin ring is reduced instead. I recognize that the following observation applies to the molecular limit, but Zn(I) is an exceedingly rare oxidation state for zinc. The observations about differences in H₂ yield for free-base versus Zn(II) porph. species are likely correct, but the explanation seems at odds with well-known chemistry.*

Response: We thank the reviewer for these valuable and professional comments! First, we agree with the reviewer that, Zn(II), whether in molecules or semiconductor materials, can hardly be reduced to low valence to work as active sites for redox reactions (e.g. H₂ evolution). In fact, our research on the use of QDs and non-noble metal elements on artificial photosynthesis (*J. Am. Chem. Soc.* **2014**, *136*, 8261; *Angew. Chem. Int. Ed.* **2017**, *56*, 3020; *Nat. Rev. Chem.* **2018**, *2*, 160) has also demonstrated that, due to the occupied d-orbitals, Zn(II) is not a good candidate for electrochemical or photochemical catalysis compared to Ni(II) and Co(II) with 3d⁸ and 3d⁷ configurations. For example, Ni(II) in the QDs/Ni(OH)₂ can facily be photo-reduced to Ni(I) and finally to Ni(0), which acts as the actual active sites of H₂ evolution confirmed by a combination of *in operando* spectroscopic techniques of electron paramagnetic resonance and X-ray absorption spectroscopy (*J. Am. Chem. Soc.* **2020**, *142*, 4680). On the contrary, the photoreduction of Zn(II) can hardly be observed, as no empty d-orbitals left to accommodate additional electrons (*Adv. Funct. Mater.* **2018**, *28*, 1801769). Therefore, Zn(II)-based systems, such as Zn-porphyrin, usually work as the chromophore unit that harvests photons to generate high-energy exciton participating in redox reactions (*Angew. Chem. Int. Ed.* **2012**, *51*, 2618; *Chem* **2018**, *4*, 1696).

In our case, the ZnPor-DETH-COF only works as the light absorbers, because no hydrogen gas is produced in the absence of external cocatalysts (Pt nanoparticles). Therefore, Zn(II) can't be reduced to low-valence species, i.e. Zn(I) or Zn(0), for

hydrogen evolution. Furthermore, X-ray absorption spectroscopy experiments have also confirmed that no low-valence Zn species are formed after light irradiation, indicating the stability of ZnPor-DETH-COF. Different from zinc(II)porphyrin molecule, the ZnPor-DETH-COF is a crystalline material, which was formed through π - π interaction of the macrocycle and the Zn \cdots Zn interaction. Under light excitation, photo-generated electron is delocalized in the whole Zn \cdots Zn chain [$Zn_n^{2n+} + e^- \rightarrow Zn_n^{(2n-1)+}$ ($n \gg 1$)] rather than localized on a specific Zn $^{2+}$ center (Jiang, D. et al *Angew. Chem. Int. Ed.* **2012**, *51*, 2618), thus avoiding the formation of concrete Zn(I) species. Subsequently, the photo-generated electrons will immediately migrate to Pt-nanoparticles bound to the surface for initiating reductive reaction. We do wish that we have made this point clear. We have modified some sentences in the revised manuscript on page 12 and 13.

Reviewer #3

Comment: *Having considered the revised manuscript and the reply to referees, I only have some comments on the reply of the authors to my comments (numbering refers to that of my original comments):*

Response: We thank the reviewer for this positive comment! We have addressed these concerns according to the suggestion in the revised manuscript.

Comment 4: *It is great that the authors have now measured an AQE value and report it in the supplementary information but they really in my opinion should also refer to it in the main manuscript. A reader shouldn't have to dig for this sort of information in the supplementary information.*

The AQE value measured is rather low (0.063%) and much lower than that of other COFs reported in the literature, e.g. TP-BDDA-COF (1.8%), FS-COF (3.2%) and N₃-COF (0.44%), making it even more important for the authors to discuss this value.

Response: We thank the reviewer for these valuable and professional comments! We find the apparent quantum efficiency (AQE) for these aforementioned COFs is obtained under different conditions. For TP-BDDA-COF (1.8% AQE), 10 mg of catalyst was dispersed in 20 mL water/TEOA (10%V) mixture and H₂PtCl₆ (3 wt% Pt). For FS-COF (3.2% AQE), 5 mg sample was suspended in an aqueous solution containing ascorbic acid (0.1 M, 8 mL) and hexachloroplatinic acid (5 μ L, 8 wt% aqueous solution). For N₃-COF (0.44% AQE), 10 mg of COF was suspended in buffer (PBS, 10 mL of 0.1 M solution at pH 7) containing TEOA (1,000 μ L; 7.38 mmol) and hexachloroplatinic acid (10 mL, 8 wt% aqueous solution). For ZnPor-DETH-COF in this manuscript, 0.50 mg sample was suspended in 2.5 mL PBS (0.1 M, pH 7) buffer containing of 10 μ L TEOA and 3.8 wt% Pt, giving an AQE is 0.063%. In fact, by taking sodium ascorbate as sacrificial reagent (reaction condition: 0.5 mg sample, 2.5 mL PBS (0.1 M, pH 7), 5 mg sodium ascorbate, 15.2 wt% Pt), AQE could be determined as 0.32%. According to reviewer's suggestions, the discussion on AQE results have been added in revised manuscript on page 9 and revised SI on page S36.

Comment 5: Fair enough, I am aware that other groups have used similar “quality” CVs to extract HOMO and LUMO values. However, to me the mere fact that other people for whom cyclic voltammetry most likely is a black box technique have done the same is perhaps not as convincing as for the authors.

Response: We thank the reviewer for these valuable comments and rigorous scientific attitudes. Indeed, since COFs are different from molecular system in morphology, crystallinity and size, the CV method might be a black box technique to measure the HOMO and LUMO values of COFs, although it has been widely used in many reported literatures. As COFs can be considered as organic semiconductors, we then performed XPS valence spectroscopy to gain the band position of MPor-DETH-COFs under feasible experimental conditions (Figure R1). Obviously, the valence-band position of these four COFs determined from XPS experiments (Table R1) generally match with the CV results. The corresponding data have been added in the revised manuscript on page 7 and SI as Supplementary Figure 20 and Table 6.

Figure R1. Valence band XPS spectra of a) H₂Por-DETH-COF, b) CoPor-DETH-COF, c) NiPor-DETH-COF, d) ZnPor-DETH-COF. The binding energy scale was calibrated using the C 1s peak at 284.60 eV.

Table R1. Comparison of the results of the determination of HOMO by CV and valence band spectra of XPS.

Samples	HOMO vs vacuum (CV)	HOMO vs NHE (CV)	XPS (valence band) vs NHE
H ₂ Por-DETH-COF	-5.60	1.10	1.22
CoPor-DETH-COF	-5.42	0.92	0.71
NiPor-DETH-COF	-5.62	1.12	1.14
ZnPor-DETH-COF	-5.70	1.20	1.26

***Comment 7:** Thanks for the explanation. I feel, however, it is being waisted in the reply to referees. I would suggest that the authors copy (part of) their reply to my comment 7 in the supplementary information into the main text.*

Response: We thank the reviewer for these positive comment. Accordingly, we have made some changes in the revised manuscript on page 11 and 12, especially the explanation of LMCT processes.

REVIEWERS' COMMENTS

Reviewer #1 (Remarks to the Author):

The authors have addressed the question raised by reviewer 2, because the ZnPor-DETH-COF only works as the light absorbers in the photocatalytic reaction. Therefore, I would like to recommend the publication in Nature Communications in its current form.

Reviewer #1:

***Comment:** The authors have addressed the question raised by reviewer 2, because the ZnPor-DETH-COF only works as the light absorbers in the photocatalytic reaction. Therefore, I would like to recommend the publication in Nature Communications in its current form.*

Response: We thank the reviewer for these very positive comments and supporting on the publication.